# Mitochondrial D-loop sequence variation and maternal lineage in the endangered Cleveland Bay horse

Andy C. Dell[1,2]*, Mark C. Curry[1], Kelly M. Yarnell[3], Gareth R. Starbuck[3], Philippe B. Wilson[2,3]*

1 Department of Biological Sciences, University of Lincoln, Lincoln, United Kingdom, 2 Rare Breeds Survival Trust, Stoneleigh, Warwickshire, United Kingdom, 3 School of Animal, Rural and Environmental Sciences, Brackenhurst Campus, Nottingham Trent University, Southwell, Nottinghamshire, United Kingdom

* andy.dell@btinternet.com (ACD); PhilippeWilson@rbst.org.uk (PBW)

**Data Availability Statement:** All relevant data are within the manuscript and on Figshare, DOI: 10.6084/m9.figshare.13154429.

## Abstract

Genetic diversity and maternal ancestry line relationships amongst a sample of 96 Cleveland Bay horses were investigated using a 479bp length of mitochondrial D-loop sequence. The analysis yielded at total of 11 haplotypes with 27 variable positions, all of which have been described in previous equine mitochondrial DNA d-loop studies. Four main haplotype clusters were present in the Cleveland Bay breed describing 89% of the total sample. This suggests that only four principal maternal ancestry lines exist in the present-day global Cleveland Bay population. Comparison of these sequences with other domestic horse haplotypes (Fig 2) shows a close association of the Cleveland Bay horse with Northern European (Clade C), Iberian (Clade A) and North African (Clade B) horse breeds. This indicates that the Cleveland Bay horse may not have evolved exclusively from the now extinct Chapman horse, as previous work as suggested. The Cleveland Bay horse remains one of only five domestic horse breeds classified as Critical on the Rare Breeds Survival Trust (UK) Watchlist and our results provide important information on the origins of this breed and represent a valuable tool for conservation purposes.

## Introduction

The Cleveland Bay is one of only five domestic horse breeds listed as critical ($<$ 300 breeding females) in the Rare Breeds Survival Trust Watchlist, making it one of Britain's oldest and most at-risk native horse breeds. The breed is thought to have been established in the seventeenth century from crosses between pack horse or "Chapman" mares, known to have been bred by monastic houses in the North-East of the UK, and newly imported hot-blooded Arabian, Barb or Mediterranean stallions, to produce an early example of what is now known as a warmblood [1]. Cleveland Bays flourished throughout the eighteenth-century achieving world renown as coaching and driving horses [2]. The Cleveland Bay Horse Society (CBHS) was formed in 1884 and the first stud book was produced with pedigrees dating back to 1732 [2]. The first stud book was officially closed in 1886 and the breed has maintained a closed stud

**Funding:** The authors received no specific funding for this work.

**Competing interests:** The authors have declared that no competing interests exist.

book since that date. However, even by the time the CBHS was formed, the breed was in decline; subsequent losses in two World Wars, and increasing mechanisation in transport and on land meant that by the middle of the twentieth century the breed was very close to extinction, with only four pure bred stallions remaining. The efforts of a few dedicated breeders, including HM the Queen, brought the Cleveland Bay horse back from the brink of extinction [3]. Recent pedigree analysis shows that only one out of 11 founder stallion lines survive in the current population, and only 9 out of 17 dam lines, of which 3 are represented in the living population by only one or two animals each [1, 2]. The average level of inbreeding for the Cleveland Bay is estimated as 22.5%, substantially higher than for the majority of domestic horse breeds (6.55–12.55%) and the current effective population size is calculated at 32; substantially lower than the United Nations FAO critical threshold of 50. Indeed, we have recently outlined a comparative analysis of the population employing both pedigree evaluation and molecular methods, whilst formulating a pedagogically-based strategy for conservation assessments of such breeds going forward and allowing for our framework to be employed for the conservation of endangered breeds going forward [4]. Furthermore, this work develops on our previous assessment of pedigree records and indicates that the small size of the population and high levels of inbreeding constitute a powerful argument for establishing a robust understanding of the genetic diversity of the breed in order to secure its survival.

The complete sequencing of the mitochondrial DNA (mtDNA) of the domestic horse was carried out by Xu and Árnason (1994). Since that time much has been done to uncover the origins of horse domestication as well as understanding matrilineal relationships within and between breeds.

Despite the increasing number of domestic horse breeds being studied, mtDNA sequencing has not previously been conducted on a representative sample of the Cleveland Bay horse. In this study, we present the mtDNA D-loop sequencing of a sample of 96 Cleveland Bay horses and describe the haplotypes found in the breed. Phylogenetic analysis of the sequence data was carried out in order to establish maternal lineages within the breed and discern relationships with other domestic horse breeds.

## Results

### Haplotype and DNA polymorphism analysis

Sequence analysis of 421 base pairs across the 96 Cleveland Bay samples identified 11 different haplotypes with 27 variable positions (Table 1). Haplotype diversity ($h$) across the sample set was determined as 0.7973 whilst nucleotide diversity ($\pi$) was 0.1537. The average number of nucleotide differences ($k$) was 7.363. Of the 11 haplotypes identified, four were matched to known female ancestry lines, by comparison of sample identity with pedigree data from the studbook database. The relationship between haplotypes with documented ancestry lines is shown in the neighbour joining tree in Fig 1, and further detailed in Table 2.

Two Cleveland Bay sequences we found to share the haplotype of the reference sequence (**X79547**). These animals (CB001 and CB003) are of Grading Registry origin and so are descended from animals that were brought into the studbook from outside the breed, being selected for reasons of pedigree or phenotype. It should be noted that these two animals are not registered in the purebred section of the Cleveland Bay studbook [2].

CB Haplotype 1 is shared by 25 individuals, representing 26% of the animals sequenced. This haplotype is shared by members of both Female Ancestry Lines 1 and 3, indicating that they are of common maternal origin, although this predates the studbook records. CB Haplotype 2 is shared by 27 of the horses sequenced, representing 28% of the sample. This haplotype is unique to animals from Female Ancestry Line 6. CB Haplotype 3 is common to 13 horses,

**Table 1. Variable nucleotides in a 497bp fragment of the mitochondrial DNA D-loop of the Cleveland Bay haplotypes compared with the reference sequence GenBank accession number X79547.**

| ID | Haplotype (Achilli) | Haplotype (Jansen) | n | Freq | 15494 | 15495 | 15496 | 15534 | 15538 | 15542 | 15584 | 15585 | 15596 | 15597 | 15600 | 15601 | 15602 | 15603 | 15617 | 15635 | 15649 | 15650 | 15659 | 15666 | 15703 | 15709 | 15720 | 15721 | 15806 | 15826 | 15827 |
|---|---|---|---|---|---|---|---|---|---|---|---|---|---|---|---|---|---|---|---|---|---|---|---|---|---|---|---|---|---|---|---|
| X79547 | A1 | A5 | 3 | 0.021 | T | T | A | C | A | C | C | G | A | A | G | T | C | T | T | C | A | A | T | G | T | C | G | C | C | A | A |
| Hap_1 | M | C1 | 25 | 0.260 | . | C | . | . | . | . | . | . | . | . | . | . | T | . | C | . | . | . | C | . | . | . | A | T | T | . | G |
| Hap_2 | G | A1 | 27 | 0.281 | . | C | . | . | . | T | . | . | . | G | . | . | T | . | . | T | . | G | . | A | C | . | A | . | . | . | . |
| Hap_3 | I | B2 | 13 | 0.135 | . | C | . | . | G | . | . | A | . | . | . | C | T | . | . | . | . | G | . | . | . | T | A | T | . | G | G |
| Hap_4 | M | B | 21 | 0.219 | . | C | . | . | . | . | . | A | . | . | A | . | T | . | . | . | . | . | . | . | . | . | A | T | T | . | . |
| Hap_5 | I | B1 | 1 | 0.010 | . | C | . | . | G | . | . | A | G | . | . | . | T | . | . | . | . | G | . | . | . | T | A | T | . | G | . |
| Hap_6 | N | C2 | 1 | 0.010 | . | C | . | . | . | . | . | A | . | . | . | C | T | . | . | . | . | . | . | . | . | . | A | T | T | . | G |
| Hap_7 | M | C | 1 | 0.010 | . | C | . | . | . | . | . | . | . | G | . | . | T | . | C | . | . | . | C | . | . | . | A | T | T | . | G |
| Hap_8 | L | D1 | 2 | 0.021 | C | C | G | T | . | . | . | . | . | . | . | . | T | C | . | . | G | . | . | . | . | . | A | T | . | . | . |
| Hap_9 | L2 | D2 | 2 | 0.021 | C | C | G | T | . | . | . | A | . | . | . | . | T | C | . | . | G | . | . | . | . | . | A | T | . | . | . |
| Hap_10 | I | B | 1 | 0.010 | . | C | . | . | G | . | T | A | . | . | . | . | T | . | . | . | . | G | . | . | . | T | A | T | . | G | . |

* sites 15585 15597 & 15650 have been previously identified as mutation hotspots [11, 18].

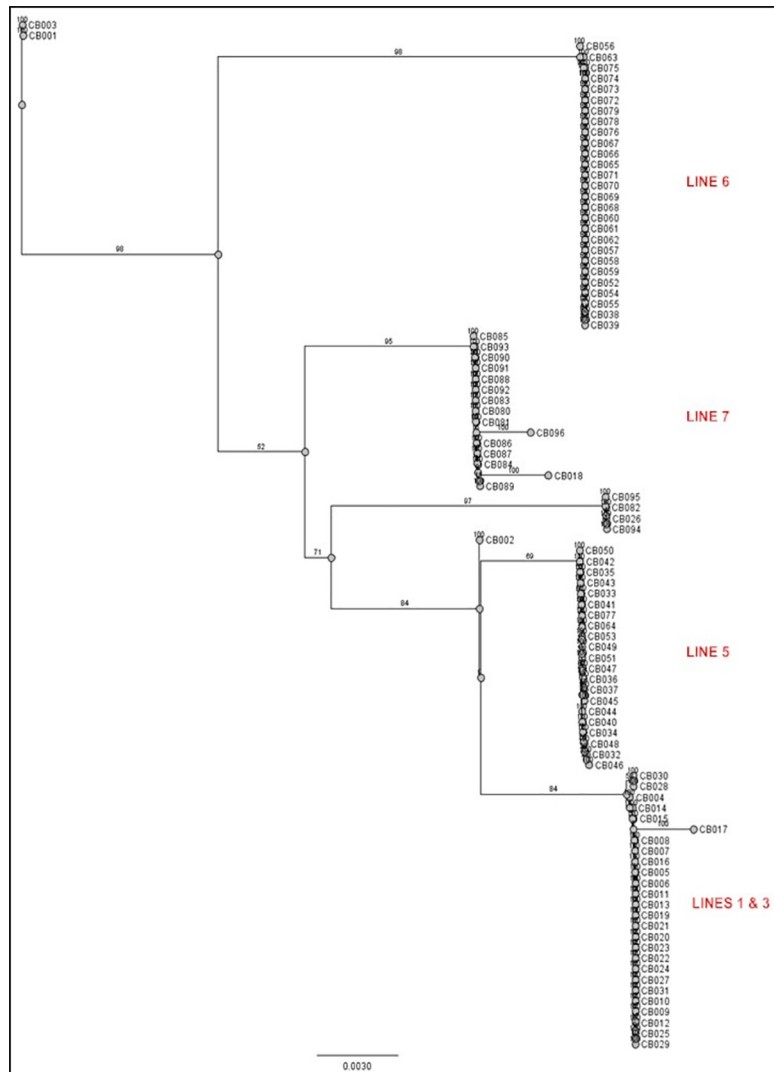

**Fig 1. Neighbour Joining tree of individual Cleveland Bay mtDNA contigs showing association of haplotype with previously described maternal ancestry lines (Emmerson 1984).** Bootstrap figures show confidence in the branches of the tree.

representing 13.5% of the sample. All of these animals have maternal origins in Female Ancestry Line 7. CB Haplotype 4 has 21 members, representing 21.9% of the sample. This haplotype is unique to members of Female Ancestry Line 5. These four Cleveland Bay haplotypes represent 89.5% of all of the animals sampled.

CB Haplotype 5 is unique to one animal (CB018) who is the only representative of Female Ancestry Line 2. CB Haplotype 6 is found in one animal (CB002); this horse has an application for Grading Register status pending with the Cleveland Bay Horse Society and so is of unconfirmed pedigree or maternal origin. CB Haplotype 7 is unique to one animal (CB017) whose pedigree places her in Female Ancestry Line 1. CB Haplotype 8 is shared by two seemingly unrelated animals (CB026 and CB082). These horses trace back to Female Ancestry Lines 3 and 7 respectively. Logically they would be expected to be of haplotypes 1 and 3. A BLAST search on the GenBank nucleotide database, for similar sequences, shows that this haplotype

**Table 2. Relationship of 11 Cleveland Bay mtDNA haplotypes to known Cleveland Bay maternal ancestry lines (total sample = 96).**

| Haplotype | n | Freq | Ancestry Line |
|-----------|----|-------|-----------------------|
| X79547 | 3 | 2.1% | Reference Sequence |
| CB Hap 1 | 25 | 26.0% | Lines 1 & 3 |
| CB Hap 2 | 27 | 28.1% | Line 6 |
| CB Hap 3 | 13 | 13.5% | Line 7 |
| CB Hap 4 | 21 | 21.9% | Line 5 |
| CB Hap 5 | 1 | 1.0% | Line 2 |
| CB Hap 6 | 1 | 1.0% | Grading Register |
| CB Hap 7 | 1 | 1.0% | Line 1 mutation |
| CB Hap 8 | 2 | 2.1% | Non Cleveland? |
| CB Hap 9 | 2 | 2.1% | Line 8 Grading Register |
| CB Hap 10 | 1 | 1.0% | Line 9 Grading Register |

equates to D1 [3], which is globally the most common of all domestic horse haplotypes. There are nine variable positions between this haplotype and the reference sequence, which is suggestive that the difference may not be down to sequencing errors. The reasons for these two seemingly unrelated animals appearing to share a non-Cleveland Bay haplotype warrants further investigation. CB Haplotype 9 is shared by two individuals (CB094 and CB095). These two animals trace back to Female Ancestry Line 8, which is a grading registry line of relatively recent origin. They are the only representatives of Line 8 in the sample. CB Haplotype 10 is unique to one animal (CB096). This horse is the only representative of the most recent female ancestry line–Curlew–identified in an earlier study [1]. Again the origins of this line trace back to the Grading Register, and to equine mtDNA Clade B [3].

## Relationships with other breeds

The four main Cleveland Bay mtDNA haplotypes produced significant matching with other domestic horse breeds. Cleveland Bay Haplotype 1 (CB Hap1) gave 100% matches in both pairwise identity and identical sites with four Kerry Bog Pony sequences [5]. There were no complete matches for CB Hap 2. However, there was 99.8% matching with sequences from Irish Draught, Arab and Akhal-Teke horses [6]. CB Hap3 was a complete match for two Irish Draught sequences [5], as well as three from Orlov horses [5]. CB Hap4 showed 99.6% identity with three Irish Draught horse sequences and one from a Zhongdian horse [7].

Of the minor haplotypes, the reference sequence [8] and two Cleveland Bay grading register animals gave 100% matches in both pairwise identity and identical sites with three Irish Draught Horse sequences and with three Kerry Bog Pony sequences [5]. CB Hap5 was a 99.6% match to four Irish Draught horse sequences as well as one of Mongolian origin [9]. Again, no exact match was found for CB Hap6, but there was >99.4% matching with Kerry Bog Pony, Irish Draught, Mongolian, and Zhongdian horse sequences. CB Hap7 was best matched at 97% similarity with five Kerry Bog Pony sequences. There were 100% matches between CB Hap 8 and Ahkal-Teke Irish Draught and Chinese Guan Mountain horses. CB Hap9 showed identity of >99.8% with 5 Irish Draught Horse sequences, whilst there was >99.8% matching between CB Hap10, Irish Draught, Kerry Bog Pony, Polish Arabian and Orlov sequences [10].

The clustering of Cleveland Bay haplotypes with other equine breeds is tabulated in Table 3 whilst a median joining network of previously defines equine clades [11] illustrating how the Cleveland Bay haplotypes fits the established model is shown in Fig 2.

**Table 3. Relationship between Cleveland Bay mtDNA haplotypes and other domestic horse breeds based on Blast searches against NCBI GenBank nucleotide database.** Clade nomenclature as defined by Jansen et al (2002).

| CB Haplotype | Female Ancestry Line | Clade | Breeds Clustered With |
|---|---|---|---|
| CB Hap 0 (Reference Sequence) | Grading Register | A5 | |
| CB Haplotype 1 | Lines 1 & 3 | C1 | Kerry Bog Pony |
| | | | Exmoor |
| | | | Icelandic |
| CB Haplotype 2 | Line 6 | A1 | Irish Draught |
| | | | Arab |
| | | | Akhal-Teke |
| | | | Danish Horse |
| CB Haplotype 3 | Line 7 | B2 | Irish Draught |
| | | | Orlov |
| | | | Arab |
| | | | Thoroughbred |
| CB Haplotype 4 | Line 5 | C | Irish Draught |
| | | | Zhongdian |
| | | | Exmoor |
| | | | Icelandic |
| CB Haplotype 5 | Line 2 / 7 | B1 | Irish Draught |
| | | | Mongolian |
| CB Haplotype 6 | Grading Register | C2 | Kerry Bog Pony |
| | | | Irish Draught |
| | | | Mongolian |
| CB Haplotye7 | Line 1 | C | Kerry Bog Pony |
| CB Haplotype 8 | - | D1 | Akhal-Teke |
| | | | Irish Draught |
| | | | Guan Mountain Horse |
| CB Haplotype 9 | Line 8 (GR) | D2 | Irish Draught |
| CB Haplotype 10 | Line 9 (GR) | B | Irish Draught |
| | | | Kerry Bog Pony |
| | | | Polish Arabian |
| | | | Orlov |

## Discussion

### Haplotype analysis

Ninety six Cleveland Bay horses were sampled and sequenced in this study. The haplotypic diversity (*h*) calculated for the breed is significantly lower than that determined for the majority of other domestic equines (*h* = 0.7973) [12]. Haplotypic diversity is indeed higher in Avar horses (*h* = 0.93), Hungarian ancient horses (*h* = 0.989), modern Akhal Teke (*h* = 0.945), [13]; Hispano-Breton heavy horse (*h* = 0.975) and Pre horse (*h* = 0.878) [14]; Lusitano (*h* = 1.0), Asturcon (*h* = 0.80), Argentine Crillo (*h* = 1.0), and Barb (*h* = 0.933) [15]. Breeds with lower haplotypic diversity include Caballo de Corro (*h* = 0.733), Paso Fino (*h* = 0.60), Florida Cracker (*h* = 0.667) and Sulphur Mustang with the lowest reported (*h* = 0.333) [15]. Each of the breeds with reported *h* lower than that found in the Cleveland Bay has been derived from significantly smaller sample sets, with *n* = 6 in each case.

Altogether 11 haplotypes were identified, 4 were found to explain 89% of samples, as well as 7 minor ones. All of the 11 haplotypes have been previously identified in domestic horses [12].

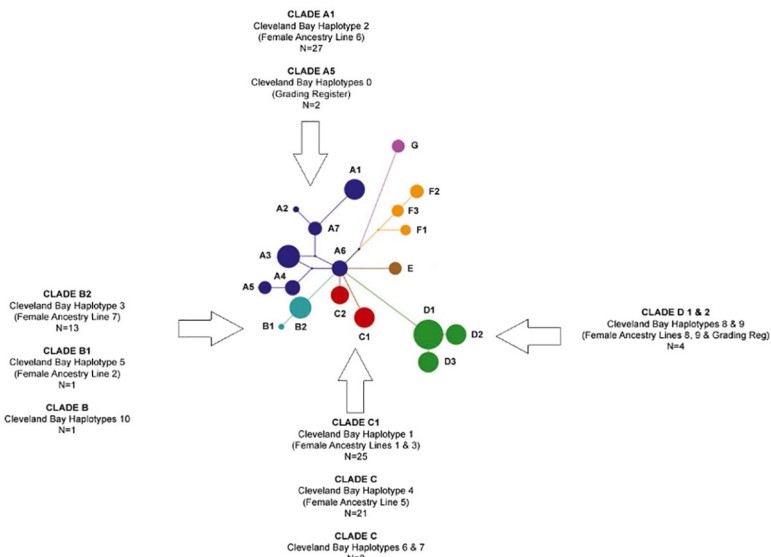

**Fig 2. Median joining network showing relationship of Cleveland Bay haplotypes to previously defined equine clades (Jansen et al 2002).** After [5].

The four principal haplotypes are associated with the following Female Ancestry lines and founders.

CB Hap1 corresponds to Line One/Three: Stainthorpe's Star (foaled circa 1850 by Grand Turk 138). This mare predates Dais(y) 318 (the previously recorded founder of Line 3 [2]) by some 26 years. As both share the same mtDNA haplotype we can consider that both lines share a common female ancestor. This haplotype was found in 26% of the samples tested, and projection from pedigree records indicates that it is present in 33% of the reference population. CB Hap2 is associated with Line Six: Trimmer 269 (foaled 1880 by Wonderful 359) and carries the unique CB Hap 2. This haplotype was found in 28% of the samples tested and, by pedigree analysis, is present in 28.6% of the reference population. This haplotype matches the previously defined type A1 [11].The only other haplotype from the same Clade found in the samples tested was CB Hap 0; shared by two Grading Register animals and the Reference sample. This equated to Jansen's type A5 [11].

CB Hap3 associates with female ancestry Line Two / Seven: Depper 39 (foaled 1855 by Ottenburgh 222). Whilst Line 2 is virtually extinct in direct descence within the reference population ($n = 3$), we demonstrate that a haplotype with significant similarity is shared with the more recent and more populated Line 7, with only one base difference at position 15597 between the haplotypes of these two ancestry lines. Whilst site 15597 is known to be a mutational hotspot [11], which could equally explain the haplotype difference; both of these haplotypes arise from cluster C, as defined by earlier work on equine mitochondrial haplotype sharing [3]. Line 7 was established by the breeding of Mr J Sunley of Gerrick House, in the 1930s, from the Grading Register mare Brilliant. This mare will have carried a haplotype which differs only in two mutations from the haplotype carried by Line 2 animals (CB Hap3). It likely that these lines have a common maternal origin, however it is difficult to deduce from the mutational differences whether the link is in recent or historic generations. Also of very similar haplotype is the recent Grading Register addition of Female Line 9 (Curlew). This has a single mutational difference with Female Line Two and belongs to the same mtDNA haplotype Clade B [11]. CB Hap3 was found in 13.5% of the samples tested and is present in 15.2% of the reference population. CB Hap4 = Line 5: Depper 42 (foaled 1880 by Barnaby 21) corresponds

to the line described by Jansen's Clade C origins [11]. 22% of the animals tested carry this haplotype, which is reflected in 20% of the reference population.

Seven minor haplotypes were identified, each being seen in either single instances or no more than two other individuals. Study of the pedigree of the horses in which these were identified indicates that the majority occur in individuals of relatively recent introduction to the breed through the Grading Register, or have *Grading Register pending* status. Others appear to be closely related to one of the major haplotypes, with one, or at most two base pair differences. The estimated rate of mutation of the equine mitochondria DNA control region is $2-4 \times 10^{-8}$ site$^{-1}$ year$^{-1}$ [16], equating to approximately one mutation per 100,000 years [17]. Several authors have identified mutational hotspots within the control region of equine mitochondrial DNA [11, 18]. Positions 15585, 15597 and 15650 are recognised as being subject to mutation at significantly greater rates than other sites [12]. Within our sample, two singleton haplotypes occur, which are unique in respect of mutations at these hotspots. CB Hap6 varies from CB Hap4 by a single mutation at 15585. Similarly, CB Hap7 varies from CB Hap1 by a single mutation, in this case at position 15597. In addition, if mutation at 15585 is ignored, then CB Haplotypes 8 and 9 appear identical.

CB Hap 8 and 9 originate from Jansen's Clade D [11]. Two animals demonstrate haplotype D1 and the remaining two haplotype D2. CB Hap 9 is associated to Female Ancestry Line 8, which corresponds to only 1% of the reference population. This ancestry line is of recent grading registry origin, tracing back to Church House Queenie GR60 by Kingmaker 1807. This suggests introgression of a female of non-Cleveland Bay origins into the breed.

Whilst studbook records dating back to the late 18$^{\text{th}}$ century suggest that as many as 17 different female founders contributed to the breed, the present study has identified only eleven haplotypes. Within these 11 haplotypes, 4 are representative of 89% of the modern day, purebred Cleveland Bay population. Accepting that the seven minor haplotypes are linked to known mutational hotspots and hence can be considered minor mutations of one of the four principal haplotypes or to relatively recent introductions into the breed *via* the Grading Register, then it is conceivable that only four female founders contribute to the modern day population. However, it is difficult to reliably deduce the timescale of this founding event or events. Indeed, these four females will certainly predate the formation of the studbook and most likely represent four different mares involved in early domestication of the horse; Jansen et al (2002) determined that horse domestication could be reasonably explained by only 77 founding females.

## Relationships with other horse breeds

Searches with the four principal Cleveland Bay haplotypes against those sequences held in the NCBI Genbank nucleotide database [19] reveal substantial haplotype sharing with the Irish Draught horse [6]. CB Haplotype 1, which is shared by Lines 1 and 3, is shown to be identical to that found in the Kerry Bog pony in Ireland. This is an old breed, closely related to the Exmoor and believed to have descended from the native pony. This supports the long held premise of these Cleveland Bay lines 1 and 3 having Chapman origins. The remaining three major haplotypes share identity with Irish Draught horse sequences [5]. The Cleveland Bay studbook is substantially older than that of the Irish Draught and it is probable that Cleveland Bay mares exerted some influence in the early breeding of working, riding and carriage horses in Ireland. Interestingly, no BLAST searches [20] found matches for any Cleveland Bay sequences with horses which could be described as being of *Carting Blood*, such as Shires, Clydesdales or the Suffolk Punch. The evidence from the the current work suggests that the carthorse has indeed played no part in the development of the Cleveland Bay breed.

The distribution of the four main Cleveland Bay haplotypes across Clades A–C [11] is consistent with the association of these Clades with horses of Northern European, Iberian or North African origin. Clade C1 has previously been associated with Exmoor, Fjord, Icelandic and Scottish Highland Ponies [11]. This cluster is geographically restricted to central Europe, the British Isles and Scandinavia, including Iceland [21, 22]. Some horses of Iberian origin have previously been associated with Clade A [11], and this is consistent with the historical records for the Cleveland Bay breed [1]. Horses of Lusitano, Pre and Sorria origins have been shown to belong to Clade B [23] whilst cluster D1 is considered as representative of Iberian and North African Breeds [11].

Traditionally the Cleveland Bay horse is thought to have evolved (maternally) from the now extinct Chapman packhorse, which in turn is considered to have originated from the native British pony, and is supported by the evidence that members of female ancestry lines one, three and five belong to Clade C. There are BLAST associations with the Exmoor and Kerry Bog Ponies, both ancient breeds, suggesting evolution from ponies that were native to post-glacial Britain. Clades A and B have associations with horses of Iberian and North African origins, respectively. The historical evidence suggests that stallions from Spain and North Africa were imported to North East England and used on local mares. It is not unlikely that good quality mares were imported from these same origins, and that these were covered with early Cleveland Bay stallions. If this is indeed the case, then mares of Line 6 and 7, and by association the almost extinct Line 2, may not be of maternal Chapman descent, but originate from Iberian and Barb mares.

## Note

The 96 contig sequences obtained in this study have been submitted to the NCBI GenBank database. The accession numbers are HQ848967 to HQ849062 inclusive.

## Materials and methods

### Population sampling

Mane hair samples were obtained from Cleveland Bay horses from Europe, North America and Australasia including the UK (78), France (5), USA (3), Canada (6), and Australia (4). Authorisation to import samples from outside of the European Union was obtained from the UK Department for Environment, Food and Rural Affairs (DEFRA) (Authorisation No. POAO/2010/238).

A total of 125 hair samples were obtained, screened, by comparison with the studbook database to determine female ancestry, with a final selection of 96 samples chosen to optimally represent the living Cleveland Bay population. Sample selection was made to ensure that a) samples were tested from each of the most critically rare female ancestry lines; and b) the remaining samples should be in proportion to the occurrence of the relevant female ancestry line in the current Cleveland Bay population according to maternal founder analysis. The final distribution of ancestry lines across the 96 samples tested is presented in Table 4.

### Ethical statement

Sampling was limited to the collection of hairs pulled from the mane or tail by the horse owner or researcher. All animal work was conducted in accordance with and approval from the international and national governing bodies at the institutions in which samples were collected (The University of Lincoln Research Ethics Committee (UREC)) and in accordance with all relevant international guidelines and frameworks including the EU PREPARE framework:

**Table 4. Female ancestry line representation in 96 Cleveland Bay hair samples selected for mtDNA d-loop sequencing.**

| Female Ancestry Line | Founder & Studbook Number | Number of Samples | Sample References | % total |
|---|---|---|---|---|
| One | Stainthorpe's Star | 14 | CB004 –CB017 | 14.58 |
| Two | Depper 39 | 1 | CB018 | 1.04 |
| Three | Daisy 318 | 13 | CB019 –CB031 | 13.54 |
| Four | Marvellous 72 | 0 | untraced | 0 |
| Five | Depper 42 | 20 | CB032 –CB051 | 20.83 |
| Six | Trimmer 268 | 28 | CB052 –CB079 | 29.17 |
| Seven | Brilliant GR | 14 | CB080 –CB093 | 14.58 |
| Eight | Church House Queenie GR60 | 2 | CB094 –CB095 | 2.08 |
| Nine | Curlew GR | 1 | CB096 | 1.04 |
| Grading Register | Various | 3 | CB001 –CB003 | 3.13 |

Smith, AJ, Clutton, RE, Lilley, E, Hansen KEAa, Brattelid, T. (2018): PREPARE: Guidelines for planning animal research and testing. Laboratory Animals, 52(2): 135–141. DOI: 10.1177/0023677217724823.

## PCR amplification and sequencing

**DNA extraction.** DNA was extracted from hair follicles using a Quaigen DNeasy 96 Blood & Tissue Kit (QIAGEN, Manchester, UK), following the Purification of Total DNA from Animal Tissues DNeasy 96 Protocol [24].

**PCR.** Polymerase chain reactions (PCR) were carried out on an MJ Research DNA Engine Tetrad PTC-225 (Bio-Rad Laboratories, Watford, UK). Forward and reverse primers were designed according to previously published work [25] for the D-loop equine Reference Sequence X79547 [8].

Primer 1 –(1F)—CGCACATTACCCTGGTCTTG

Primer 2 - (1R)–GAACCAGATGCCAGGTATAG

PCR amplification of mtDNA was carried out in a 96 well microtitre plate, with each well containing 5μl Primer 1 (10nM), 5μl Primer 2 (10nM), and 10 μl Qiagen HotStarTaq Plus Master Mix (QUIAGEN, Manchester, UK) and 1μl template.

The PCR reaction took place under the following thermal cycling sequence: The reaction mixture was heated to 95˚C for 5 minutes followed by 30 cycles of 94˚C for 40 seconds; 55˚C for 45 seconds; 72˚C for 45 seconds. Thermocycling concluded with extension at 72˚C for 10 minutes following which the product was held at 12˚C. Following the completion of the PCR reaction, all PCR products were cleaned using a Zymo Research ZR-96 DNA Clean & Concentrator™-5 (Zymo Research, Irvine, California, USA). Samples were eluted in 50μl of DNA free water.

**Sequencing.** Sequencing of PCR products was carried out using Big Dye™ Terminator Cycle Sequencing Kit v 3.1 (Applied Biosystems Inc., Foster City, California, USA). The sequencing reaction comprised 3μl sequencing primer (forward or reverse) (3.2nM), 3μl sequencing template (purified PCR product) and 4μl Big Dye terminator v 3.1 (Applied Biosystems Inc., Foster City, California, USA). Thermocycling conditions: 95˚C for 5 minutes followed by 25 cycles of 96˚C for 10 seconds, 50˚C for 5 seconds, 60˚C for 3 minutes with a final extension at 72˚C for 10 minutes, subsequently maintained at 12˚C prior to sequencing. All products of the sequencing PCR were cleaned *via* passage through individual Sephadex clean up columns (Sigma-Aldrich, Gillingham, UK), to remove any unincorporated dye terminator products and diluted with 10μl of DNA free water. Sequences were generated on an Applied

Biosystems Inc. 3730xl 96 capillary DNA analyser (Applied Biosystems Inc., Foster City, California, USA).

## Data analyses

The forward and reverse AB1 sequence files for each sample were assembled into contigs using the software Geneious v 4.8 [26]. All 96 contigs were then aligned using the equine mtDNA d-loop reference sequence (GenBank accession number X79547 [8]), with the Geneious v 4.8 software [26]. Haplotype and DNA polymorphism analyses was conducted using DNAsp [27].

## Relationships with other breeds

To consider haplotype sharing with other equine breeds, Basic Local Alignment Search Tool (BLAST) queries with each of the haplotypes identified in this study were conducted against the GenBank nucleotide database using Geneious [26]. In order to further, understand genetic relationships between Cleveland Bay horses and other domestic equines, the 11 haplotypes identified were compared with the sequence motifs and clades described by Jansen et al (2002) (Table 3).

## Acknowledgments

Mitochondrial DNA D-loop sequencing was carried out at the Dublin laboratory of Source Bioscience Ltd, who were contracted to undertake the processes of extraction, amplification, PCR and sequencing.

## Author Contributions

**Conceptualization:** Andy C. Dell, Mark C. Curry.

**Data curation:** Andy C. Dell, Mark C. Curry.

**Formal analysis:** Andy C. Dell, Kelly M. Yarnell, Philippe B. Wilson.

**Funding acquisition:** Mark C. Curry.

**Investigation:** Andy C. Dell.

**Methodology:** Mark C. Curry, Gareth R. Starbuck, Philippe B. Wilson.

**Project administration:** Kelly M. Yarnell, Gareth R. Starbuck, Philippe B. Wilson.

**Resources:** Kelly M. Yarnell, Gareth R. Starbuck, Philippe B. Wilson.

**Supervision:** Philippe B. Wilson.

**Writing – original draft:** Andy C. Dell.

**Writing – review & editing:** Kelly M. Yarnell, Gareth R. Starbuck, Philippe B. Wilson.

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
