## [Decision Letter · Decision Letter 0]

13 Oct 2020

PONE-D-20-24140

Mitochondrial D-loop sequence variation and maternal lineage in the endangered Cleveland Bay horse

PLOS ONE

Dear Dr. Wilson,

Thank you for submitting your manuscript to PLOS ONE. After careful consideration, we feel that it has merit but does not fully meet PLOS ONE’s publication criteria as it currently stands. Therefore, we invite you to submit a revised version of the manuscript that addresses the points raised during the review process.

Both reviewers found it to be sound and well written. Some of the material is very horse and breed specific and this was the reason for differing opinions between the reviewers. I do have to agree with reviewer two that it would have been better to have combined your two papers on this study to provide one very robust manuscript. Please look at the nomenclature suggestion provided by reviewer one and provide a revised manuscript.

We look forward to receiving your revised manuscript.

Kind regards,

Chris Rogers

Academic Editor

PLOS ONE

Additional Editor Comments:

Thank you for your submission. Both reviewers found it to be sound and well written. Some of the material is very horse and breed specific and this was the reason for differing opinions between the reviewers. I do have to agree with reviewer two that it would have been better to have combined your two papers on this study to provide one very robust manuscript. Please look at the nomenclature suggestion provided by reviewer one and provide a revised manuscript.

Journal Requirements:

"University of Lincoln and The Rare Breeds Survival Trust"

"No"

"No"

Reviewers' comments:

Reviewer's Responses to Questions

**Comments to the Author**

1. Is the manuscript technically sound, and do the data support the conclusions?

Reviewer #1: Yes

Reviewer #2: Yes

2. Has the statistical analysis been performed appropriately and rigorously? 

Reviewer #1: Yes

Reviewer #2: Yes

3. Have the authors made all data underlying the findings in their manuscript fully available?

Reviewer #1: Yes

Reviewer #2: Yes

4. Is the manuscript presented in an intelligible fashion and written in standard English?

Reviewer #1: Yes

Reviewer #2: Yes

5. Review Comments to the Author

Reviewer #1: This paper presents analysis of mtDNA typing of the Cleveland Bay horse, one of the world's most endangered horse breeds. The paper is well written and appropriate methods are used. I have only a couple of comments. One, the mtDNA haplotype nomenclature of Jansen has largely been replaced by that of Achilli et al. 2012. Using Jansen is ok but Achilli should also be used concurrently. The diversity of mtDNA in the CB is consistent with the low nuclear DNA variation that has recently been reported and the high inbreeding level within the breed. It is interesting how well the maternal lines predicted by pedigree analysis are aligned with the mtDNA haplotypes. I also am somewhat surprised by how high the proportion of haplotypes representing horses from the British Isles is. Within breed ancestry representations are often quite diverse representing past intermixing of horses prior to breed formation. I also will say that the diversity shown does not indicate that the Chapman horse is not the direct link to all parts of the CB ancestry as there is no knowledge of the make-up of that horse.

My one negative comment is not actually a criticism of the paper. This paper has a great deal of very specific information about the make-up of the individuals that comprise the breed. This has almost no meaning to most potential readers but is valuable information for the specialist. The great detail of the specific maternal lines and groups within the breed make certain sections of this paper very difficult to follow. In fact the paper presents little that is new except in reference to the breed. The paper might be more appropriate for a different journal although I don't have a specific recommendation for one.

Reviewer #2: The current study aimed to establish maternal lineages within the Cleveland Bay Horse breed and detect relationships with other domestic horse breeds. The Cleveland Bay Horse is a very important breed and known to have low genetic diversity. The authors used typical methods for the data analysis of part of the D-loop. I think sequencing the full D-loop would be more informative especially with the high level of similarity seen among tested samples. Although the results and discussion were well written, the approach and results are lacking novelty, very specific to horses and not applicable for novel discoveries in other species. The overall outcomes of the current study might be interesting and useful for a narrow group of readers. More importantly, I wish the authors have included the finding of this study with their previous paper. The story could be then much informative by integrating the autosomal genetics findings in the light of the mtDNA sequencing instead of having them separated in two manuscripts.

6. PLOS authors have the option to publish the peer review history of their article (what does this mean?). If published, this will include your full peer review and any attached files.

Reviewer #1: No

Reviewer #2: No

---

## [Author Response · Author response to Decision Letter 0]

30 Oct 2020

Reviewer #1: This paper presents analysis of mtDNA typing of the Cleveland Bay horse, one of the world's most endangered horse breeds. The paper is well written and appropriate methods are used. I have only a couple of comments. One, the mtDNA haplotype nomenclature of Jansen has largely been replaced by that of Achilli et al. 2012. Using Jansen is ok but Achilli should also be used concurrently. 

The nomenclature from Achilli has been incorporated within Table 2.

The diversity of mtDNA in the CB is consistent with the low nuclear DNA variation that has recently been reported and the high inbreeding level within the breed. It is interesting how well the maternal lines predicted by pedigree analysis are aligned with the mtDNA haplotypes. I also am somewhat surprised by how high the proportion of haplotypes representing horses from the British Isles is. Within breed ancestry representations are often quite diverse representing past intermixing of horses prior to breed formation. I also will say that the diversity shown does not indicate that the Chapman horse is not the direct link to all parts of the CB ancestry as there is no knowledge of the make-up of that horse.

My one negative comment is not actually a criticism of the paper. This paper has a great deal of very specific information about the make-up of the individuals that comprise the breed. This has almost no meaning to most potential readers but is valuable information for the specialist. The great detail of the specific maternal lines and groups within the breed make certain sections of this paper very difficult to follow. In fact the paper presents little that is new except in reference to the breed. The paper might be more appropriate for a different journal although I don't have a specific recommendation for one.

Reviewer #2: The current study aimed to establish maternal lineages within the Cleveland Bay Horse breed and detect relationships with other domestic horse breeds. The Cleveland Bay Horse is a very important breed and known to have low genetic diversity. The authors used typical methods for the data analysis of part of the D-loop. I think sequencing the full D-loop would be more informative especially with the high level of similarity seen among tested samples. Although the results and discussion were well written, the approach and results are lacking novelty, very specific to horses and not applicable for novel discoveries in other species. The overall outcomes of the current study might be interesting and useful for a narrow group of readers. More importantly, I wish the authors have included the finding of this study with their previous paper. The story could be then much informative by integrating the autosomal genetics findings in the light of the mtDNA sequencing instead of having them separated in two manuscripts.

We thank the reviewers for their comments, and their careful assessment of the manuscript.

---

## [Editor Report · Decision Letter 1]

10 Nov 2020

PONE-D-20-24140R1

Mitochondrial D-loop sequence variation and maternal lineage in the endangered Cleveland Bay horse

PLOS ONE

Dear Dr. Wilson,

Thank you for submitting your manuscript to PLOS ONE. After careful consideration, we feel that it has merit but does not fully meet PLOS ONE’s publication criteria as it currently stands. Therefore, we invite you to submit a revised version of the manuscript that addresses the points raised during the review process.

Please look at the revised manuscript in the results section. In the copy uploaded there are a number of occurrences of the following Error! Reference source not found

We look forward to receiving your revised manuscript.

Kind regards,

Chris Rogers

Academic Editor

PLOS ONE

Additional Editor Comments (if provided):

Please look at the revised manuscript in the results section. In the copy uploaded there are a number of occurrences of the following Error! Reference source not found

Please correct and upload a new version.

---

## [Author Response · Author response to Decision Letter 1]

10 Nov 2020

Dear Colleagues,

Further to our recent submission, we have modified the main manuscript to refer to our first paper concerning the genetic analysis of the Cleveland Bay horse population through pedigree evaluations and molecular methods. Furthermore, we have reiterated the tutorial-based nature of our initial paper, which guides the reader through our framework and proposes its implementation in the decision-making process for conservation of endangered breeds and species. Whilst we recognise the points of the Editorial Office in terms of segmenting research projects, the authors would like to firmly stress that these pieces of work are not only standalone, but represent logical and careful developments of the narrative as part of the large project we continue to undertake in conserving the Cleveland Bay horse and the native equine breeds of the United Kingdom. Whilst we use these breeds as examples herein, our approaches are careful to be extendable not just to equines and livestock, but in wider conservation practices.

Furthermore, according to the email dated 10th November 2020, we have amended the manuscript to remove the “Error! Reference source not found” therein.

Updates to statements based on the requests from Editorial Office:

1) Thank you for updating your data availability statement. You note that your data are available within the Supporting Information files, but no such files have been included with your submission. At this time we ask that you please upload your minimal data set as a Supporting Information file, or to a public repository such as Figshare or Dryad. 

Please also ensure that when you upload your file you include separate captions for your supplementary files at the end of your manuscript.

As soon as you confirm the location of the data underlying your findings, we will be able to proceed with the review of your submission.

Authors: These data have been provided: 10.6084/m9.figshare.13154429. The statements have also been amended.

2) Your ethics statement should only appear in the Methods section of your manuscript. If your ethics statement is written in any section besides the Methods, please move it to the Methods section and delete it from any other section. Please ensure that your ethics statement is included in your manuscript, as the ethics statement entered into the online submission form will not be published alongside your manuscript. 

Authors: This has been moved.

3) It is important that you include a cover letter with your manuscript. Please ensure that this letter is addressed specifically to PLoS ONE. Please also include

* why this manuscript is suitable for publication in PLoS ONE.

* how does your paper provide a worthwhile addition to the scientific literature?

* how does your paper relate to previously published work? 

* which types of scientists do you believe will be most interested in your study?

Authors: This serves as the document.

4) Thank you for stating the following financial disclosure: 

NO

Authors: The authors received no specific funding for this work.

5) Thank you for stating the following in your Competing Interests section: 

"No"

Please complete your Competing Interests on the online submission form to state any Competing Interests. If you have no competing interests, please state "The authors have declared that no competing interests exist.", as detailed online in our guide for authors at https://eur03.safelinks.protection.outlook.com/?url=http%3A%2F%2Fjournals.plos.org%2Fplosone%2Fs%2Fsubmit-now&data=04%7C01%7Cphilippe.wilson%40ntu.ac.uk%7Cb7c2ef235df54ce631fe08d87b5be85f%7C8acbc2c5c8ed42c78169ba438a0dbe2f%7C0%7C0%7C637394981962863769%7CUnknown%7CTWFpbGZsb3d8eyJWIjoiMC4wLjAwMDAiLCJQIjoiV2luMzIiLCJBTiI6Ik1haWwiLCJXVCI6Mn0%3D%7C1000&sdata=ex8IGrShDq%2B7e39fAIsGyALzEK6Z7TDaR0HTS5u2E4c%3D&reserved=0

Authors: The authors have declared that no competing interests exist

Original letter:

Herein, we consider the genetic diversity within the Cleveland Bay horse population, one of the oldest and most foundationally integral equine breeds. Indeed, until now, there was little convincing evidence concerning the loss of genetic variation within the breed, across the generations, leading to a status as endangered as that today. We therefore approach the problem of rigorously developing breed management plans for a critically endangered population such as the Cleveland Bay horse by first considering the genealogical and molecular data pertaining to the generational genetic history of the breed.

The Cleveland Bay horse is a historic breed of horse native to the UK which has been widely used throughout the equine world for generations. However, since the move to mechanization, its uses on land, in the field became limited and the population threatened. Now critically endangered, the breed must be subject to strict management based on genetic factors in order to increase diversity therein and safeguard the Cleveland Bay for future generations. Herein, we demonstrate that only four original female lines are represented in the current population, and consider the evolution of the breed from the now-extinct Chapman Horse. We present the methodology as a powerful tool for conservation, both livestock and in applications to wild species in situ. Furthermore, we dispel previous assumptions regarding the lineage and ancestry of the modern Cleveland Bay population.

Whilst being evidently pertinent to livestock and native breed conservation biology, our approach is inherently translatable to in situ conservation of species more globally. Indeed, our follow-up paper on this work, develops this study and describes 16 years of breed management programmes, and their effect on the genetic diversity within the population; work which could not have taken place without the robust and necessary first principles established within the present study.

In addition to a robustly-researched and thoroughly described methodology, we present the logical development of the work and fundamental theory throughout in order for this work to be recognised as a valuable standalone resource for biological conservationists.

We suggest the following colleagues as reviewers for this manuscript:

Prof Stephen Hall: stHall@lincoln.ac.uk

Prof Tim Morris: Tim.Morris@nottingham.ac.uk

Dr Tim Bray: tbray@bristolzoo.org.uk

We hope you will consider this manuscript with the attention it deserves, and look forward to receiving your feedback.

With kind regards,

Prof Philippe B. Wilson

MChem(Hons) PhD PGCertHE MRSC FRSB FLS FHEA

---

## [Editor Report · Decision Letter 2]

18 Nov 2020

Mitochondrial D-loop sequence variation and maternal lineage in the endangered Cleveland Bay horse

PONE-D-20-24140R2

Dear Dr. Wilson,

We’re pleased to inform you that your manuscript has been judged scientifically suitable for publication and will be formally accepted for publication once it meets all outstanding technical requirements.

Kind regards,

Chris Rogers

Academic Editor

PLOS ONE
---

## [Editor Report · Acceptance letter]

20 Nov 2020

PONE-D-20-24140R2 

Mitochondrial D-loop sequence variation and maternal lineage in the endangered Cleveland Bay horse 

Dear Dr. Wilson:

I'm pleased to inform you that your manuscript has been deemed suitable for publication in PLOS ONE. Congratulations! Your manuscript is now with our production department. 

Kind regards, 

on behalf of

Dr. Chris Rogers 

Academic Editor

PLOS ONE